# Implementation of Artificial Intelligence in Diagnostic Practice as a Next Step after Going Digital: The UMC Utrecht Perspective

**DOI:** 10.3390/diagnostics12051042

**Published:** 2022-04-21

**Authors:** Rachel N. Flach, Nina L. Fransen, Andreas F. P. Sonnen, Tri Q. Nguyen, Gerben E. Breimer, Mitko Veta, Nikolas Stathonikos, Carmen van Dooijeweert, Paul J. van Diest

**Affiliations:** 1Department of Pathology, University Medical Center Utrecht, 3508 GA Utrecht, The Netherlands; r.n.flach-2@umcutrecht.nl (R.N.F.); n.l.fransen@umcutrecht.nl (N.L.F.); a.f.p.sonnen-3@umcutrecht.nl (A.F.P.S.); t.q.nguyen@umcutrecht.nl (T.Q.N.); g.e.breimer-2@umcutrecht.nl (G.E.B.); mitko.veta@gmail.com (M.V.); nstatho2@umcutrecht.nl (N.S.); c.vandooijeweert-3@umcutrecht.nl (C.v.D.); 2Department of Biomedical Engineering, Eindhoven University of Technology, 5600 MB Eindhoven, The Netherlands

**Keywords:** artificial intelligence, machine learning, digital pathology, roadmap, implementation

## Abstract

Building on a growing number of pathology labs having a full digital infrastructure for pathology diagnostics, there is a growing interest in implementing artificial intelligence (AI) algorithms for diagnostic purposes. This article provides an overview of the current status of the digital pathology infrastructure at the University Medical Center Utrecht and our roadmap for implementing AI algorithms in the next few years.

## 1. Background

In 2007, we started with the first implementation of a digital pathology system, initially by building up a digital archive for quick revision of cases for and in support of multidisciplinary team meetings, research, and teaching [1]. For scanning roughly 137,000 histological stains and 30,000 immunohistochemical (IHC) stains annually, at that time, we acquired three Aperio ScanScope XT scanners that provided the desired capacity of 700 slides per day. Images acquired at 20× were stored in proprietary pyramid multiresolution.svs file format in a resolution of 0.50 µm/pixel. After the diagnostic process was finished in the traditional microscopic way, slides were scanned. At the time, no quality control of the whole slide images (WSI) was performed. Only scanning failures that were seen by chance were manually corrected. Making use of the vendor’s application programming interface (API) and software development kit (SDK), we were able to integrate with our pathology reporting system and laboratory information system.

As to storage, the first iteration was a hierarchical storage management solution (Sun Microsystems, Santa Clara, CA, USA). Initially, all images were stored on fiber channel hard disk drives for rapid access and also copied to a scalable tape library in a buffered way. This storage hardware remained in place until migration to an all object based storage disk system. Because of performance problems of the first iteration of the all disk storage system [1], we migrated to the new hospital-wide disk-based bulk storage system with a superb performance.

The first generation of our digital pathology system started to show signs of aging by the end of 2014. Scanning capacity was no longer sufficient because of our growing practice, so we decided to go for a completely new setup to enable fully digital diagnostics, which was implemented in 2015. It comprised three high throughput Hamamatsu XR scanners and one Hamamatsu RS scanner (Hamamatsu City, Japan) for fluorescence and big slides, and the Sectra Picture Archiving Communication System (PACS) (Sectra AB, Linkoping, Sweden).

The system has performed adequately for 6 years, signing out >95% of our histology cases digitally. Occasionally, we revert back to slides for pediatric pathology, mitoses, microorganisms recognition, birefringence assessments, and hematopathology. We do cytology still with the microscope because of a lack of scanning capacity and storage for Z-stack scanning, since this would result in a lack of confidence in digital diagnostics with current image quality. We have seen several important developments in the PACS, such as the implementation of tools to support mitoses and Ki67 counting, a bidirectional link between our reporting system and the PACS, placeholder thumbnails for stains requested and for lacking images. Also, patient safety increased by magnification-sensitive tracking of our movements through the slides to preclude missing tissue parts and flagging thumbnails of unreviewed slides.

## 2. Current Setup and Activities

We have recently renewed the contracts with Sectra and Visiopharm (the reseller of Hamamatsu in The Netherlands) and have migrated to a new single pathology reporting system with LIMS (Delphic AP, Sysmex, New Zealand). In 2022, we expect to incorporate two regional pathology laboratories into our digital pathology infrastructure. The recent versions of Sectra PACS and Delphic AP are ready to function as a multicenter digital pathology workflow system, which will allow us to work as one virtual team of fully superspecialized pathologists over three locations. In addition, we will be installing four NanoZoomer S360 Hamamatsu high throughput scanners and one NanoZoomer S60 Hamamatsu scanner for fluorescence and whole mounts.

In 2022, dedicated cytology whole slide scanners are expected to enter the market, which we hope to evaluate and purchase to make the jump to digital cytology, without seriously impacting storage.

## 3. AI Implementation: Current Status and Road Map

At UMC Utrecht, we aspire to implement AI as much and as soon as possible, thereby unleashing the full potential of digital pathology, with benefits for both patients and pathologists. Various studies on AI-implementation, both prospective and retrospective, are currently ongoing within the UMC Utrecht. Examples are the CONFIDENT trials, which will be discussed below. Several algorithms are available that have been developed through collaborations with the Radboud University in Nijmegen and the Technical University of Eindhoven, The Netherlands, that are ready for further testing and validation in daily practice [2,3,4]. Besides, we work with several companies bringing AI algorithms to the market on implementation. We expect to make pathology diagnostics more objective, faster and intellectually more satisfying, while more importantly our patients will also benefit from the best tissue diagnostics that forms the basis for personalized treatment.

Pathology has always been a medical specialty that was in the frontline of automation (e.g., electronic reporting, speech recognition, image analysis, structured reporting). Although lagged several decades behind radiology in going digital, this was largely due to lack of affordable and fast scanners and infrastructure to handle big image files. There is at this moment a big wave in pathology to catch up with going digital, and we expect AI to be adopted fairly organically. Likely, in view of our inclination towards automation and use of computers, pathologists will easily learn to use and interpret AI interactively, so probably not much education will be necessary. This does not take away that using AI should be user-friendly and integrated into PACS systems [5].

Our Sectra PACS includes an algorithm for assessing the percentage of Ki67 positive nuclei, which is based on AI. Further, we have integrated an in-house developed AI algorithm for recognizing mitotic figures. In an interactive way, an area of interest can within the PACS be demarcated on the WSI after which the algorithm finds mitoses and mitosis-like objects and displays them in galleries. Objects can easily be moved between these galleries to arrive at a final AI-assisted mitotic count (Figure 1).

At this moment, we are evaluating Qualitopix, a new stain quality control algorithm from Visiopharm, and Derm-AI, a Proscia algorithm for workflow stratification of dermatopathology cases. Within the framework of our new contract with Visiopharm, we will soon implement their breast cancer AI package, consisting of algorithms for ER, PR, HER2, Ki67, and lymph node metastases. We aim to run these algorithms entirely in the background so results will be ready when the pathologist opens up the case.

## 4. Developing AI-Implementation Studies

AI algorithms might be implemented in various ways, depending on the algorithm. Some algorithms can be used solely for workflow optimization; for example, for identifying cases that do not need additional diagnostics, or assigning difficult cases to expert pathologists [6]. It might also improve tumor grading consistency [2,7,8]. Whereas currently most AI validating studies are designed retrospectively, useful prospective trials are currently lacking [9].

The design of prospective studies is based on the interests of the many parties involved in AI-implementation in daily clinical practice. First, patients need an accurate diagnosis. For example, no tumor cells may be missed, and tumors must be graded accurately and consistently. While the former is currently achieved in daily clinical practice by using IHC stainings in all negative cases, the latter is not. Significant inter- and intra-laboratory variation in grading of various tumor types (colorectal, breast, prostate) has been observed nationwide [10,11,12,13,14]. As grade can be decisive in treatment choice, the pathologist is pivotal in guiding treatment of cancer patients, and consistency is warranted [15,16]. [Flach, under review] Here, AI algorithms may help pathologists grade more accurately and consistently, and might even serve as a second ‘reviewer’.

From a pathologist’s point of view, in a field with an ever growing workload, searching, for example, for tumor metastases in (sentinel) lymph nodes is a time-consuming task. It requires meticulous assessment of slides, in general with an overall low yield. Therefore, looking diligently may not be compelling, and pathologists may be prone to use IHC stainings in most, if not all cases, thereby putting pressure on the budget of the pathology department. AI assistance of pathologists on this task may not only save on IHC, but it may lower pathologists’ workload, as it has been shown that AI-assisted grading is less time consuming than traditional grading [8].

From the department’s financial point of view, costs of the growing number of IHC stainings sometimes even exceed the compensation for assessment of the complete resection specimen. Calculations from our hospital showed, for example, that we spent over €13,000 to detect nine cases of lymph node metastases in 95 sentinel nodes from 68 breast cancer patients. The majority of these (6/9) were not even deemed clinically relevant by medical oncologists, who consider isolated tumor cells in patients without neoadjuvant treatment irrelevant in relation to treatment strategy [17].

In cervical cancer, IHC identified only three patients with micrometastasis and five patients with isolated tumor cells undetected with H&E staining in 630 sentinel nodes from 234 patients. To achieve this, 3791 slides were stained with IHC at an estimated additional cost of €94,775. In 1.4% (95% CI 0.3–4.3%) of patients, routine use of IHC adjusted the adjuvant treatment [18].

For prostate cancer, performing IHC staining as standard of care is not necessarily advised when carcinoma is obviously present or absent [19]. However, it does help pathologists identify small foci, the extent of the tumor and can assist in tumor grading, which is critical in prostate cancer risk stratification and decision-making for performing pelvic lymph node dissection [20]. For this purpose, we spent €22,000 on triple p63/CK5/AMACR IHC staining in a 3-month period in 27 cases.

This financial point of view has to be considered when assessing the viability of business cases for digital pathology and AI implementation. A complex matter, as digital pathology is often seen as an ‘add-on’, as it does not replace the physical slides, which also need to be kept and stored, at least for now. AI, however, may tip this balance to the side of benefit as it has the potential to improve cancer grading and reproducibility, thereby improving patient treatment and potentially outcome, while lowering costs. This is specifically promising, as the current trend in oncology seems to be that improving patient care may only be realized at higher costs [21].

Lastly, from a legal perspective, algorithms for clinical use must be certified (FDA-approved or IVDR-approved). Currently, the first algorithms are reaching this stage, enabling pathologists to implement and evaluate them in prospective trials (see also below). Nevertheless, it was presumed too big a step to implement them without a safety net (for example, IHC-stainings) in the first implementation phase.

Another imperative ethical point to raise, is that it is currently unimaginable that AI-algorithms will diagnose cases unsupervised or communicate results without human input. Therefore, previous studies evaluating and comparing independent AI-algorithms to pathologists may seem nice, but situations simulated in these studies are highly unlikely to be implemented in current daily clinical practice. Therefore, we strongly feel that the aim is augmented intelligence, rather than AI independently, since pathologists and AI together have been shown to outperform either one alone [5,8,22]. For example, it has been shown that scoring of HER2 IHC staining intensity (which is relevant for treatment decision in breast cancer patients) is done more accurately by a pathologist using an AI assisted digital microscope tool compared to a non-AI assisted pathologist [23]. This is also illustrated by an international survey amongst 718 pathologists in dermatopathology, that showed that only 6% of the pathologists feared that the human pathologist would be replaced by AI in the foreseeable future. The vast majority agreed that AI will improve dermatopathology, while most of these pathologists did not have any experience with AI in their daily practice [24].

Overall, the hope is that AI will improve the quality of diagnosis, reduce the workload of pathologist’s performing these diagnostics, and reduce costs of the entire diagnostic process. However, as pointed out by Van der Laak et al., the hope is still to be distinguished from the hype in prospective trials [9].

## 5. Challenges in Trial Designs

A major challenge in prospective implementation trials is implementing a reference standard in the workflow. Here, it is essential to distinguish assessing biomarkers or other factors, for which currently no reference standard is implemented (like histologic grading or scoring percentages of cells), from tumor detection, for which a reference standard is in place, such as using IHC stainings in all negative cases [17,19].

## 6. Confident Trials

At the UMC Utrecht, we are currently running two prospective trials on clinical implementation of AI-assisted tumor detection in digital pathology (CONFIDENT). The first is the CONFIDENT B-trial which evaluates the detection of sentinel lymph node metastases in breast cancer. The second is the CONFIDENT-P trial, which evaluates tumor detection in prostate cancer. These studies aim to safely introduce an AI-assisted workflow, and should be easy to use for other algorithms in pathology practice as well. Within these prospective CONFIDENT trials, we investigate the value of AI-assistance in tumor detection in pathology specimens in the current pathology workflow.

## 7. Interactive vs. Background Processing

There are basically two forms of deployment for AI algorithms in clinical practice: on-demand and background batch analysis. The former approach is interactive, fulfilling the need of the pathologist when encountering a situation during diagnostics (Figure 2). The advantage of this approach is that analysis can be limited to relevant areas in relevant slides selected by the pathologist. The disadvantage is that, depending on the model, runtime might be long, especially if the selected area is too large. Also, the biased nature of interactively selecting certain areas in specific slides (e.g., for mitoses counting) can be considered a disadvantage. Therefore, running algorithms in the background that process full WSI may be the default approach for deploying AI models in practice (Figure 3). It is imperative that results are ready by the time the pathologist opens up the case. However, implementing such automatic processes is not trivial from a technical and functional perspective.

In order to trigger an AI system to start analysis on a WSI, it will have to either have some well-defined criteria to analyze a case, which means that well defined metadata of a case or advanced text mining of grossing description will be used to start the analysis. In absence of such information, the alternative is to perform the analysis on all possible WSI that might fit a broader selection criteria to ensure that the pathologist has access to the results. That would require an extensive hardware infrastructure to ensure that there is no latency between the time the case is ready and the time that the results are ready.

## 8. Hardware Issues

Running AI algorithms requires significant computing, especially when processing entire WSI, which are easily 10 Gigapixels. Installing and maintaining a local GPU server cluster for AI purposes at a pathology department is costly and, most of the time, an overkill since the GPU capacity will need to accommodate peak loads. This means that using an existing hospital GPU cluster or a cloud solution would be necessary. However, external cloud solutions can be a security and privacy concern. Analyzing WSI entails transferring data outside of the hospital firewall which would either have to be anonymized prior to export or the connection to cloud solution would have to be over a VPN. In addition, the security issues related to anonymizing and exporting images outside the firewall and importing AI algorithm output are not trivial, but can probably be solved.

## 9. Certification Issues

Historically, healthcare may not be in the frontline of implementing technology tools that have already transformed other areas of commerce and daily life [25]. One factor, among others, that hampers the implementation of new technology tools in health care is the regulation that accompanies medical products. With the promising developments in AI software technology that will assist pathologists in making a more accurate diagnosis, pathologists will in the future increasingly depend on software technology to make their diagnosis. Implementing such AI software tools in clinical practice will improve diagnosis accuracy and therapy response prediction. Therefore, the development and implementation of these tools must not be hampered by unnecessary regulation.

However, these software tools will process sensitive personal medical data, and therefore regulation on the use of this data is necessary to prevent unconsented and secondary use of personal data. In May 2021, the new European regulation on software as a medical device (Medical Device Regulation, MDR) came into effect. This regulation changed the definition of software as a medical device and the risk classification of software. AI software tools that will help pathologists make a more accurate diagnosis now fall in a higher risk score and must be assessed by an officially appointed organization [26]. The MDR aims to improve the regulation and safety of the software used for diagnosing and treating patients. The GDPR (General Data Protection Regulation) from the European Union reduces the obligations regarding administrative formalities before accessing health data. They aim to make data actors more accountable rather than restricting their ability to develop new tools in the first place [27]. The FDA also proposes that the regulation of software development and design for health care needs a different approach than the traditional regulation of hardware-based medical devices [25]. They have therefore proposed a software pre-certification program where they assess organizations that perform high-quality software design, testing, and monitoring. The FDA program aims to develop effective medical device software, drive faster innovation, and enable timely patient access while keeping pragmatic and least burdensome regulatory oversight to verify the continued safety and performance of software tools in the real world [25]. To date, several companies have obtained CE-IVD, IVDR, or FDA approval of their algorithms. For locally developed algorithms, thorough local validation will probably be required in many countries.

## 10. Deployment of Models in Clinical Practice

The development and training of AI models that can reach decent performance has become increasingly easier in practice thanks to frameworks released by major companies like Google and Facebook (PyTorch and Tensorflow) [28,29] as well as libraries like FastAI, which offer tools to rapidly train new models in a matter of days [30]. However, despite the rapid development tools and resources available, the deployment of such models have proven much more challenging in practice. Apart from the regulatory framework needed to validate a model for clinical practice, the effort required to develop a model into a full-fledged product is a multiple of the effort to train the model. In order to effectively deploy a model in production, there has to be:The necessary infrastructure to retrain the model if and when performance drops.Records of data versions used with every version of the model released.Monitoring infrastructure.Serving infrastructure—infrastructure needed to deploy the model.

The AI field is rapidly developing, which means that the technology developed around it is also developing with the same rate. Top-performing models dating from 2 years ago, will be outdated today and will have suffered from model drift. Computer vision models trained on a first generation platform (for example Tensorflow v1), would be almost impossible to port to the latest version without redeveloping/rewriting. That rapid development, which has served as a boom for AI proliferation, has brought along long standing issues found in the rapid software development community namely technical debt [31].

Another issue in deploying AI-models in practice, is trust of the application of AI models. Recently, a lot of discussion and efforts have gone into the topic of explainable AI for medical image analysis. Explainability methods are seen as a tool that can enable or increase the transparency of AI models thus addressing some of the ethical and regulatory concerns of their use [32]. Ghassemi et al. have recently expressed scepticism about state-of-the-art explainability methods and argued that more effort should be put toward proper validation of AI methodology [33]. We generally agree with this sentiment and see explainability methods as just another tool in the toolbox of AI development and validation methods.

## 11. The Business Case

For patients, implementation of AI algorithms might result in an improved diagnostic process. However, Ho et al. already stated that digital pathology is not likely to be implemented, unless a viable business case is presented, as digital pathology diagnostics workflow comes with significant costs [34]. Next to high acquisition costs, also additional histopathology, IT personnel and costs for integrating with other medical devices and system raise costs, which laboratories cannot easily afford without external help, especially when considering future developments outlined below [34]. Ho et al. found that improving speed and quality of pathology diagnostics, which is necessary for digital pathology, comes with significant savings elsewhere in the healthcare system. The same holds for AI implementation. However, Ho et al. made their financial projections for digital pathology implementation in an integrated health care organization, serving as both a health care provider and the payor [34]. In organizations where this is not the case, it is challenging to turn budget silos into communicating vessels, so it will mostly be the pathology labs themselves that need to build a business case for AI implementation. Bluntly, time savings will likely make pathologists go home earlier, but those will rarely be on such a scale that fewer pathologist FTEs will suffice. Therefore, tangible, straightforward cost savings associated with some key AI algorithms will have to pave the budgetary way for larger-scale AI implementation. For instance, the Visiopharm company claims that their HER2 IHC algorithm reduces the 2+ category, comprising about 20% of breast cancer cases and for which expensive reflex FISH testing is indicated, by some 75%, which would amount to saving €3600 per 100 random breast cancer cases. Second, a prostate cancer algorithm facilitating finding cancer spots may obviate the need for the expensive triple p63/CK5/AMACR IHC staining, besides saving much time with regard to measurements and grading. Third, an AI algorithm that finds micro metastases and isolated tumor cells in sentinel nodes may obviate the need for cytokeratin IHC on step sections, saving up to €100 per sentinel node.

## 12. AI 2.0

With our experiences in implementing a fully digital pathology workflow, including the first AI algorithms used in daily practice, where do we see AI in pathology going in the future? Considering the current rise of genetic and proteomic methods in pathology diagnostics and the development of spatially-resolved molecular imaging modalities, i.e., spatial transcriptomics and spatial proteomics, it becomes evident that advanced machine learning algorithms will play a key role in making sense of the ever growing amount of data. Especially in the context of precision medicine in a personalized care setting, leveraging on the full potential of all data available is of the utmost importance to select the proper treatment for each patient and prevent unwanted treatments, thus saving overtreatment for the patient, and costs for society. Again, as detailed in the example of the introduction of digital pathology and AI in the UMC Utrecht, careful and stepwise introduction of algorithms will be needed in the future for both quality control and financial reasons. The following years we will see a rise in research that will try to stratify patient and treatment options based on models that include classical histology, IHC, DNA- and RNA sequencing in bulk, and spatially-resolved molecular imaging methods. Models that will be generated will rely on tabular data (sequencing) and potentially multiscale image data, making an integration and assessment of classifiers without machine learning algorithms unlikely [35,36].

However, as with digital pathology itself, the basis will initially be a well-organized data infrastructure/repository for tabular and image data on which the algorithms can work. In a modest step towards digital pathology 2.0/AI 2.0 at the UMC Utrecht, we are working towards integrating (spatially)-resolved proteomics into our diagnostics routines. We use matrix assisted laser desorption/ionization-based mass spectrometry imaging (MALDI-MSI) in various research projects using patient tissues. MALDI-MSI can provide a molecular profile of thousands of molecules at each image pixel without the loss of tissue architecture. This opens the way, for example, to assess molecular tumor heterogeneity or to look at amyloid composition together with classical histology on the same image, by carefully selecting peaks from the measured mass spectra [37]. Integrating these data into our digital pathology environment/PACS system seems natural, as pathologists are already used to annotating different regions for diagnostics. Eventually, AI algorithms will annotate regions of interest and, from these regions, pick peaks on the mass spectrum to assess molecular composition. As this example shows, there are many hows, buts, and ifs associated with such projects, ranging from file/data framework issues to acceptance by pathologists [36]. However, as our “road-trip” from fully glass-based pathology to “fully-digital” pathology at the UMC Utrecht shows, early investment into the future eventually pays off, and we believe that multiscale integration of molecular and image data—pathomics—is the future of pathology.

## Figures and Tables

**Figure 1 diagnostics-12-01042-f001:**
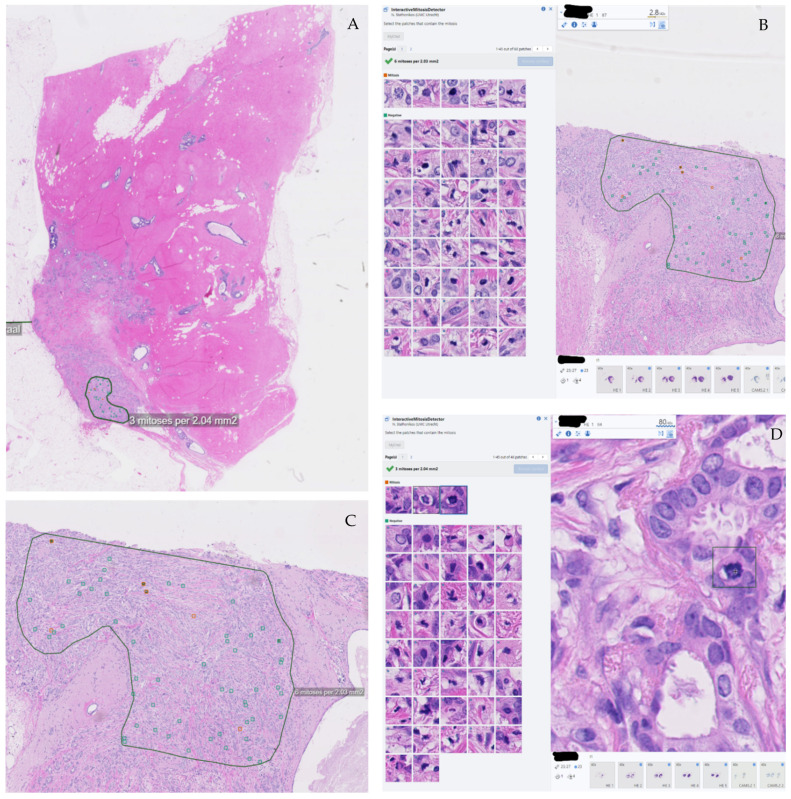
In-house developed AI algorithm for mitotic figures recognition. (**A**) Selecting a region of interest. (**B**,**C**) Interactive Mitosis Detector, with gallery (**B**) and without gallery (**C**). The detector highlights those areas suspicious for mitosis with orange, those negative for mitosis as green. (**D**) Close-up of mitotic figure (mitotic figure selected by the pointer on the right in the gallery), recognized by the algorithm.

**Figure 2 diagnostics-12-01042-f002:**

Flowchart showing a workflow for on-demand, interactive processing.

**Figure 3 diagnostics-12-01042-f003:**
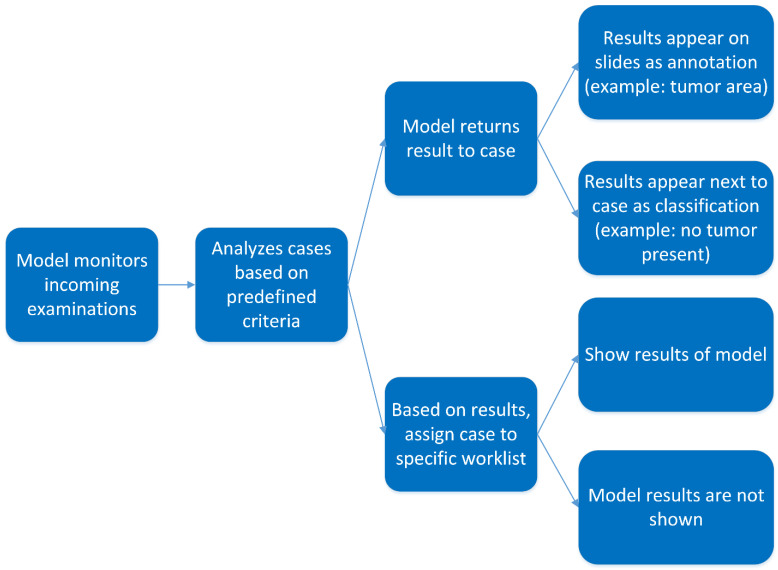
Flowchart showing workflow for background batch analysis, a workflow driven process.

## Data Availability

Not applicable.

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
