# Peer review of "Implementation of Artificial Intelligence in Diagnostic Practice as a Next Step after Going Digital: The UMC Utrecht Perspective"

_diagnostics, 2022, doi:10.3390/diagnostics12051042_

Round 1
Reviewer 1 Report
The paper is very interesting and well written.
Some minor details:
1 - in line 75 please correct "a soon as"
2 - it would be interesting so read some comment or opinion about explainability of AI, a current hot-topic (an important issue when considering the use of tools that execute some task that the pathologist is not able to perform and, as such, not able to visually confirm if the output of the algorithm is correct or not)
3 - it would be also interesting to know what the authors think about the necessary computacional education for medical users (in this case, how much computational science do pathologists need to know?)
4 - Regarding the "Bussiness Case" - if costs are so high and "laboratories cannot easily afford without external help", why are the first addopters usually the private sector (more profit oriented)
5 - line 302 - "... the Visiopharm company claims that their HER2 IHC algorithm reduces the 2+ category...."
Does this algorithm follow the current ASCO/CAP guideline? or it was trainned to predict the likelihood of the ISH result regardless of the staining pattern? Because there are alternative systems to the ASCO/CAP guideline that claim to reduce the 2+ category. In this case, what should be our approach to these new AI systems that disrupt current methodologies?
Author Response
We thank the reviewer for reviewing our article. Below we will answer all questions and suggestions.
1 - in line 75 please correct "a soon as".
Sorry, this has been corrected.
2 - it would be interesting so read some comment or opinion about explainability of AI, a current hot-topic (an important issue when considering the use of tools that execute some task that the pathologist is not able to perform and, as such, not able to visually confirm if the output of the algorithm is correct or not)
We agree, and have therefore added some sentences on explainability of AI in the paragraph ‘Deployment of models in clinical practice’, page 16, line 1-8. In addition, we included 2 references to go with this.
3 - it would be also interesting to know what the authors think about the necessary computational education for medical users (in this case, how much computational science do pathologists need to know?)
This is indeed interesting. Pathology has always been a medical specialty that was within the frontline of automation (e.g. electronic reporting, speech recognition, image analysis, structured reporting). Although lagged several decades behind radiology in going digital, this was largely due to lack of affordable and fast scanners and infrastructure to handle out big image files. There is at this moment a big wave in pathology to catch up with going digital, and we expect AI to be adopted fairly organically. Likely, in view of our inclination towards automation and use of computers, pathologists will easily learn to use and interpret AI interactively, so probably not much education will be necessary. This does not take away that using AI should be user-friendly and integrated into PACS systems. We have added these sentences to the paragraph ‘AI implementation: current status and road map’, page 5 line 5-13. We have included a reference on this topic.
4 - Regarding the "Business Case" - if costs are so high and "laboratories cannot easily afford without external help", why are the first adopters usually the private sector (more profit oriented)
We are not sure that the private pathology sector are the early adapters. So far, we see especially academic labs spearheading AI implementation. This does not take away that also for private pathology labs AI could be very important since saving pathologists' time means saving money, since time is money there.
5 - line 302 - "... the Visiopharm company claims that their HER2 IHC algorithm reduces the 2+ category...."
Does this algorithm follow the current ASCO/CAP guideline? or it was trained to predict the likelihood of the ISH result regardless of the staining pattern? Because there are alternative systems to the ASCO/CAP guideline that claim to reduce the 2+ category. In this case, what should be our approach to these new AI systems that disrupt current methodologies?
The website of Visiopharm does not state how exactly the algorithm was trained, it only states that the output of the HER2 algorithm is: Conversion of the connectivity into the classical HER2 Score of 0, 1+, 2+ or 3+ in agreement with the clinical scoring guidelines from ASCO/CAP to aid diagnosis. Therefore, no current methodologies seem to be disrupted.
Reviewer 2 Report
Even if the paper is addressed to a particular Center and even if the theme of diagnostic is faced only in the dedical field, it is important to include somethink about the mathod used (fuzzy ...AI intelligence ?.....mixed method ?).
I strongly invite the author to remark this aspect and to evaluate the possivility to discuss about neuro fuzzy sytems for diagnostical system.
I suggest to takke the following reference and to cite it.
In fact the references must be increased.
Prediction models for the corrosion phenomena in Pulp & Paper plant Autori M Bucolo, L Fortuna, M Nelke, A Rizzo, T Sciacca Data pubblicazione 2002/2/1 Pubblicazione Control Engineering Practice Volume 10 Numero 2 Pagine 227-237 Editore PergamonAuthor Response
We thank the reviewer for reviewing our article. Below we will answer all questions and suggestions.
- Even if the paper is addressed to a particular Center and even if the theme of diagnostic is faced only in the medical field, it is important to include something about the method used (fuzzy ...AI intelligence ?.....mixed method ?). I strongly invite the author to remark this aspect and to evaluate the positivity to discuss about neuro fuzzy sytems for diagnostical system.
We agree that this is interesting, but our paper is not a review about AI methodology, but focusses on implementation in daily pathology practice. Several good review papers on AI methodology have already appeared, so this is beyond the scope of our paper. - I suggest to take the following reference and to cite it. In fact the references must be increased.
We have indeed increased the number of references in view of the comments by Reviewer 1 on explainability of AI. The suggested reference is on Prediction models for the corrosion phenomena in Pulp & Paper, we assume that is a mistake.